# Interaction of the Coupled Effects of Irrigation Mode and Nitrogen Fertilizer Format on Tomato Production

Yuan Huang [1,2], Ying-Ru Yang [1,3], Jing-Xin Yu [4] , Jia-Xuan Huang [1,2], Yi-Fan Kang [1,5], Ya-Ru Du [1,2] and Guo-Ying Tian [1,5,*]

1. Shijiazhuang Academy of Agricultural and Forestry Sciences, Shijiazhuang 050041, China; 17731175112@189.cn (Y.H.)
2. Key Laboratory for Agricultural Information Perception and Intelligent Control of Shijiazhuang, Shijiazhuang 050041, China
3. Shijiazhuang Agricultural Information Engineering Technology Research Center, Shijiazhuang 050041, China
4. Intelligent Equipment Research Center, Beijing Academy of Agriculture and Forestry Sciences, Beijing 100097, China
5. Hebei Province City Agriculture Technology Innovation Centers, Shijiazhuang 050041, China
* Correspondence: tguoying1@163.com; Tel.: +86-133-2304-6816

**Abstract:** The production efficiency and quality of tomatoes is affected by the mode of irrigation and the nitrogen forms. This study explored the impacts of different irrigation regimes, nitrogen forms, and their coupled effects on tomato production. The various irrigation regimes were set at 50%FC~90%FC (W1), 60%FC~90%FC (W2), 70%FC~90%FC (W3), and 80%FC~90%FC (W4) Furthermore, the control (CK) group followed a conventional drip irrigation regime in the local area. Nitrogen forms in this study comprised urea-based fertilizer (urea N 32%, F1), nitrate-based fertilizer (calcium ammonium nitrate N 15%, F2), and ammonium-based fertilizer (ammonium sulfate N 21%, F3). Combining these two factors yielded 15 treatment groups. The experiment was conducted in a solar greenhouse, and the soil type was sandy loam soil. The research focused on observing the yield, quality, and water–fertilizer use efficiency of tomatoes under these 15 treatment groups. The results demonstrate that irrigation had a more significant impact on the yield and nutrient accumulation rate compared to the nitrogen forms. To comprehensively evaluate the yield, quality, and water–fertilizer use efficiency of tomatoes, a combination evaluation method was employed. W3F2 produced the highest yield, CKF2 achieved the highest comprehensive quality score, and W2F2 had the highest comprehensive water and fertilizer use efficiency score. Using the fuzzy Borda model, the evaluation information of the three dimensions was combined. W3F2 ranked first, suggesting the adoption of an irrigation control regime of 70%FC to 90%FC, along with the application of nitrate-based nitrogen fertilizer during the fruit set to the harvest stage. It presented the best performance of tomato yield, quality, and water–fertilizer use efficiency across multiple dimensions.

**Keywords:** facility tomato; controlled irrigation; field capacity; moving averages; comprehensive evaluation; fuzzy Borda model

## 1. Introduction

Tomato is one of the most popular fruit and vegetable crops worldwide, and in recent years, growers around the world have achieved higher yields and superior taste through improved cultivation techniques [1]. Water and fertilizer control are two important factors related to tomato yield and quality, but blindly increasing the amount of irrigation and fertilizer could lead to lower water and fertilizer use efficiency, environmental pollution, and nitrate accumulation [2]. Therefore, using irrigation sensing and control devices, such as soil moisture sensors, is an effective way to improve tomato yield, quality, and water–fertilizer use efficiency. In addition, it can improve the understanding of crop water and fertilizer requirements [3].

Sensor-based irrigation strategy, which can monitor soil content in real-time and determine when irrigation should be turned on or off, could exert significant effects on tomato production. It is generally believed that higher soil moisture content can increase crop yield [4]. Meanwhile, the reduction of irrigation can promote the accumulation of the reducing sugar, total acid, vitamin C (Vc), and total soluble solids (TSS) in fruit [5]. This effect is further impacted by other factors, particularly nitrogen fertilization. Nitrogen is the most important nutrient affecting crop growth and development, and it is a component of many essential metabolites.

There is a growing body of research on the response of various crops, such as rice [6], tobacco [7], potato [8], and lettuce [9], to different nitrogen forms. These studies have indicated that various crops exhibit differences in utilizing different nitrogen forms. Tomato, being a typical nitrate-loving crop with high nitrogen fertilizer requirements, has received considerable attention in this field. However, previous studies have primarily focused on single-factor trials or quantitative evaluations of water and fertilizer [10,11]. For example, deficit irrigation is a well-recognized water-saving strategy, achieved by the deliberate application of a sub-optimal amount of water, which may or may not result in some yield reduction. It does not enhance growth and yield as such but modifies plants' physiological processes. It can influence the efficiency of water use, which would lead to considerable water saving [12]. According to the existing research, for tomato cultivation, the application of N30-70 (30% as a basal fertilizer and 70% as a topdressing) presents higher tomato yield, lycopene and vitamin C contents, and sugar–acid ratio in fruits, but it has lower organic acid content in fruits [13].

Additionally, the interaction between different nitrogen forms and irrigation has been noted in the literature [14,15]. For example, Chu et al. showed that the application of alternate partial root-zone irrigation (APRI) should consider the soil moisture conditions combined with the appropriate nitrogen forms, and tomato plants supplied with ammonium nitrogen at the flowering period grew better than those supplied with nitrate nitrogen [16]. However, the interaction effect still requires a comprehensive evaluation, such as that regarding the fruit yield, quality, and other indicators, especially ecological benefit indicators, such as water–fertilizer utilization efficiency, rather than merely the crop growth rate.

The tomato planting effect is a comprehensive concept that includes not only the fruit yield and quality directly related to economic benefits, but also water and fertilizer utilization efficiency related to ecological benefits. It is the sum of the interactions between different individual attributes [17]. The fruit quality is also generally classified as the taste (total soluble solids, sugar, and acid) and nutritional value (lycopene and vitamin C) [18]. Water and fertilizer utilization efficiency include the productivity of water consumption and the partial factor productivity of fertilizer application [19]. When evaluating the effects of irrigation and fertilization systems on tomato production, a comprehensive evaluation should be conducted based on three aspects: tomato yield, quality, and water–fertilizer use efficiency. A comprehensive evaluation of multiple indicators generally involves three steps: indicator selection, weight determination, and scoring evaluation [20]. There are various methods for determining the weights and evaluation scores, including a hierarchical analysis based on operations research [21], factor analysis based on multivariate statistics [22], neural networks based on computer science, and algorithms that integrate multiple evaluation methods [23]. However, it is important to note that different evaluation methods may produce different results due to varying arithmetic principles. In this regard, the fuzzy Borda method can effectively synthesize single evaluations from multiple dimensions by utilizing both the evaluation value and ranking value information [24].

The primary objectives of this study are listed as follows: (1) Investigate the effects of irrigation, nitrogen forms, and their interaction on tomato leaf growth, yield, quality, and water–nutrient use efficiency. This will clarify the different forms and interactive effects of irrigation and nitrogen on various aspects of tomato growth. (2) Determine the optimal irrigation and fertilizer system and establish a pluralistic evaluation system for tomatoes.

This was accomplished by utilizing principal component analysis to evaluate the effects of tomato quality and water–fertilizer use efficiency. Additionally, the fuzzy Borda method was employed to perform a comprehensive evaluation of yield, quality, and water and fertilizer use efficiency.

## 2. Materials and Methods

### 2.1. Overview of the Test Site and Test Materials

The test was conducted at the Modern Agricultural Park of Zhao County, Hebei, China (114°78′ E, 37°76′ N). The solar greenhouse utilized in the experiment was a steel-framed structure, with a length of 90 m, a width of 8 m, and a top height of 4.5 m. The back wall was 0.5 m thick and covered with polyethylene film. The planting area was 489 m$^2$. Sandy loam soil was used in the test site, and the nutrient conditions of the soil from 0 to 80 cm depth are shown in Table 1. The test material "Seminis 313" (Seminis Seeds (Beijing) Co., Ltd., Beijing, China) is an infinite growth variety with large fruits, medium to early maturity, and resistance to tomato mosaic virus, gray leaf spot, and yellowing leaf curl virus. On 15 March 2022, tomato seedlings were grown until they had five leaves and a heart and were 10–12 cm tall. The test was conducted from the beginning of the flowering and fruiting period (15 April 2022) to the end of the harvest period (20 June 2022), with a growth period of 66 days.

**Table 1.** Soil nutrient contents.

| Depth (cm) | Organic Matter (g/kg) | Available Nitrogen (mg/kg) | Available Phosphorus (mg/kg) | Available Potassium (mg/kg) | pH | Electrical Conductivity, EC (uS/cm) |
|---|---|---|---|---|---|---|
| 0–20 | 13.50 | 54.83 | 21.50 | 179 | 7.72 | 177.20 |
| 20–40 | 16.60 | 68.32 | 17.90 | 188 | 7.81 | 179.80 |
| 40–60 | 12.00 | 57.34 | 26.80 | 172 | 7.74 | 246.20 |
| 60–80 | 11.60 | 56.53 | 19.30 | 196 | 7.86 | 232.20 |

### 2.2. Experimental Design

The experiment was designed with two factors: irrigation systems and nitrogen forms. According to the preliminary experiments of our group and the conclusions of existing studies, maintaining the soil moisture content at a 70–90% field capacity (FC), during the flowering and fruiting period, can result in good tomato growth. Consequently, the irrigation section of this experiment was designed based on various ranges of soil moisture content, and independent soil moisture sensors were installed for each treatment to keep it constant. The irrigation limits were set at 50%FC~90%FC (W1), 60%FC~90%FC (W2), 70%FC~90%FC (W3), and 80%FC~90%FC (W4) [25].

The irrigation scheme of the control treatment (CK) was formulated according to the commonly used local irrigation method and the instruction of water-soluble fertilizer. Immediately after transplanting, 120–150 m$^3$/hm$^3$ of water was applied. Then, 5–7 days post-transplanting, 120–150 m$^3$/hm$^3$ of water was applied for seedling establishment. Approximately 45–50 days after transplanting, at the fruit expansion stage, 140–170 m$^3$/hm$^3$ of water was applied per cycle every 10–15 days, totaling four cycles. Around 100–105 days post-transplanting, at the harvest stage, an irrigation cycle was applied with 10–12 m$^3$/hm$^3$ of water, and the cycle was set at 9–12 days, totaling five cycles. All forms of nitrogen fertilizer were from China Stanley Agricultural Company. Amide nitrogen fertilizer (urea with a nitrogen content of 32%, F1), nitrate nitrogen fertilizer (Ca(NO$_3$)2-4H$_2$O with a nitrogen content of 15%, F2), and ammonium nitrogen fertilizer ((NH4)$_2$SO$_4$ with a nitrogen content of 21%, F3) were used. A total of 15 treatments were established in the cross-combination of irrigation and fertilization. During the experiment, the total nitrogen amount for each treatment was kept consistent. Various nitrogen fertilizer forms were applied for each treatment in five equal portions. All forms of nitrogen fertilization would be administered through

the drip irrigation system, without foliar spraying. The fertilization for all treatments was ensured to be carried out within one week. The experiment design is presented in Table 2, which provides a clear and concise overview of the nutrient application strategy and the experimental conditions.

**Table 2.** Experimental design.

| Irrigation Level | Soil Moisture Content When Irrigation Started | Soil Moisture Content When Irrigation Stopped | Nitrogen Fertilizer Treatment | Amount of Nitrogen Applied in Different Forms/kg·hm$^{-2}$ |
|---|---|---|---|---|
| W1 | 50%FC | 90%FC | F1 | Urinary ammonia nitrogen 610.31 |
| | | | F2 | Calcium nitrate 1302 |
| | | | F3 | Ammonium sulfate 930 |
| W2 | 60%FC | 90%FC | F1 | Urinary ammonia nitrogen 610.31 |
| | | | F2 | Calcium nitrate 1302 |
| | | | F3 | Ammonium sulfate 930 |
| W3 | 70%FC | 90%FC | F1 | Urinary ammonia nitrogen 610.31 |
| | | | F2 | Calcium nitrate 1302 |
| | | | F3 | Ammonium sulfate 930 |
| W4 | 80% FC | 90%FC | F1 | Urinary ammonia nitrogen 610.31 |
| | | | F2 | Calcium nitrate 1302 |
| | | | F3 | Ammonium sulfate 930 |
| CK | Commonly used local drip irrigation system | | F1 | Urinary ammonia nitrogen 610.31 |
| | | | F2 | Calcium nitrate 1302 |
| | | | F3 | Ammonium sulfate 930 |

### 2.3. Experimental Method

The plot size was 3 m × 4.8 m, with 2 protective rows left between each plot. Raised-bed and wide-row cultivation was adopted, with a trapezoidal cross-section. Its ridge height was 20 cm, its bottom width was 70 cm, and its top width was 60 cm. Tomatoes were planted on the ridge surface, with a row spacing of 50 cm and a plant spacing of 30 cm. The observation path between the furrows was 90 cm wide. Drip irrigation was used, with an emitter spacing of 30 cm. One drip tape was placed for each row of tomatoes. A schematic diagram is shown in Figure 1. Each treatment had 64 plants, each of which was repeated three times, and all repetitions were studied separately nine times. Figure 2 shows the different stages of tomato production.

To determine the soil field capacity in the tomato cultivation experiment, soil moisture multi-profile three-dimensional monitoring equipment (Agricore Technology, Beijing, China) was utilized. The calculation method employed was moving averages. The equipment was buried in the middle ridge of each plot, at a distance of 15 cm from the plant in the north-south direction. In the test area, the soil 0~40 cm deep was initially found to be supersaturated prior to planting. Continuous data of the soil water content were gathered from the 0 to 40 cm depth layer one day before irrigation. The soil water content data were smoothed by the sliding average method with a 4-h time interval.

The sliding average of the soil moisture content data was analyzed to identify the inflection point during the receding process. Specifically, when the change rate between the current time point and the previous 4th time point did not exceed 0.4%, the 4-h average soil moisture content of the soil layer measured at the current time point and the previous three time points was considered the field capacity of the soil layer.

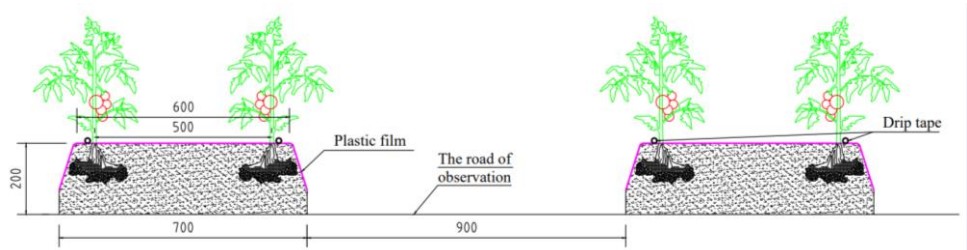

**Figure 1.** Schematic diagram of planting. Its ridge height was 200 mm, its bottom width was 700 mm, and its top width was 600 mm. Tomatoes were planted on the ridge surface, with a row spacing of 500 mm.

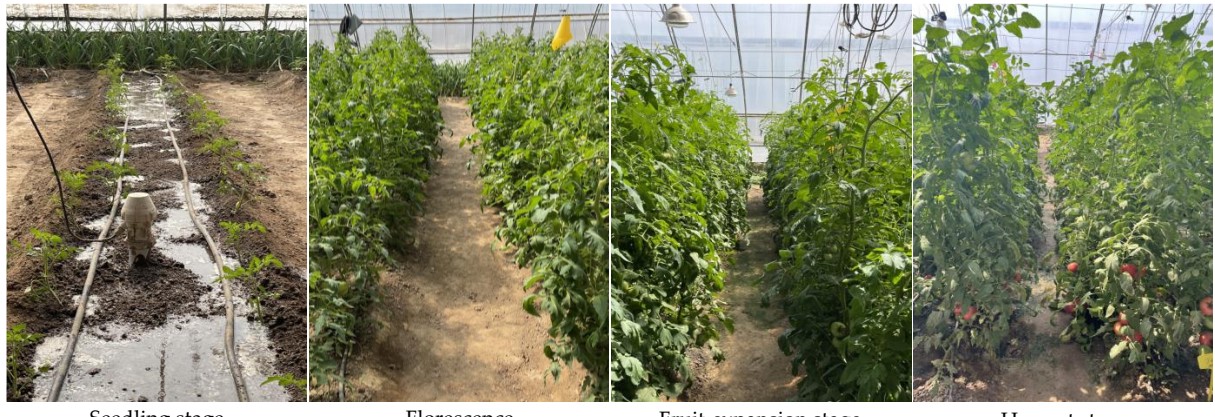

**Figure 2.** Tomato growth stages.

Undisturbed soil samples were collected using a ring knife at the same soil layer. The collected soil samples were then brought back to the laboratory, saturated with water, and placed on dried soil to allow the latter to absorb the gravitational water in the undisturbed soil. Next, 15 g of undisturbed soil was placed into an aluminum box with a constant weight ($m_0$) and immediately weighed ($m_1$). The soil sample was then dried at 105 °C to a constant weight, after which it was weighed again ($m_2$). The field capacity X of the soil layer was calculated using Equation (1) based on the measurements of the undisturbed soil samples. The procedure was performed three times with the average value of the results being reported.

$$X = \frac{(m_1 - m_2) \times 1000}{m_2 - m_0} \tag{1}$$

The results of the two field capacity measurement methods are presented in Table 3. The relative error between the measurements was within ±2%, indicating that the soil moisture multi-profile three-dimensional monitoring equipment was capable of accurately and automatically monitoring the soil field capacity.

**Table 3.** The soil field capacity of each plot.

| Irrigation Level | Field Capacity Determined Using a Ring Knife (%) | Field Capacity Determined by the Monitoring Equipment (%) | Comparison of Results | Relative Error (%) |
|---|---|---|---|---|
| W1 | 39.19 ± 1.74 [a] | 39.31 | −0.12 | −0.31 |
| W2 | 37.01 ± 1.34 [a] | 36.90 | 0.11 | 0.30 |
| W3 | 41.16 ± 1.33 [a] | 41.97 | −0.81 | −1.97 |
| W4 | 37.88 ± 2.05 [a] | 37.80 | 0.08 | 0.21 |

Note(s): Values represent mean ± standard deviation. Different lowercase letters a indicate significant differences among treatments ($p < 0.05$).

During the experiment, nitrogen fertilizer was applied in each plot, and 589.95 kg/hm$^2$ of agricultural potassium sulfate with a potassium content of 50% was applied uniformly to ensure consistent nutrient application amounts throughout the entire growth period. The experiment design is presented in Figure 3, which provides a clear and concise overview of the experimental design.

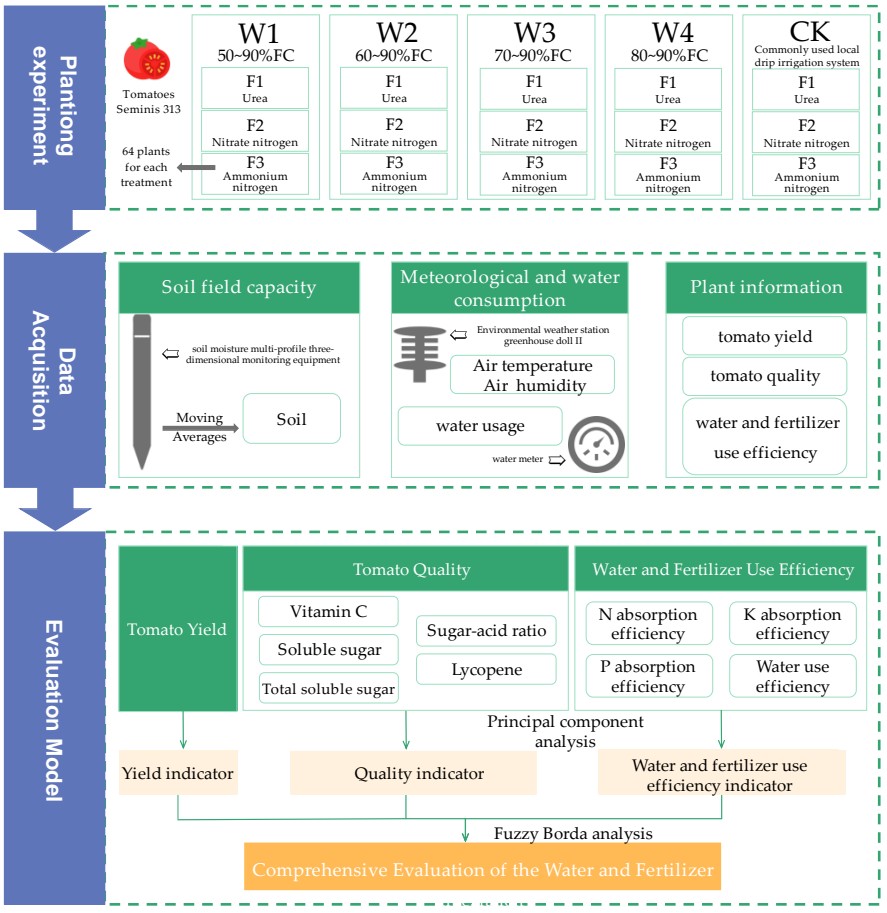

**Figure 3.** Experimental design.

*2.4. Project Measurement and Methods*

2.4.1. Solar Greenhouse Environmental Conditions and Water Consumption

The hourly air temperature and humidity were measured using a Greenhouse Doll II (Agricore Technology, Beijing, China), and the data for environmental conditions during the experiment are presented in Figure 4. Throughout the entire growth period, the air temperature ranged from 8.57 °C to 41.22 °C, and the air humidity ranged from 11.9% to 95.86%.

To monitor the water used for irrigation during the entire growth period, water meters were installed in each treatment, and the amount of irrigation water used (in m$^3$) was recorded. The amount of irrigation water used per tomato plant (in m$^3$/plant) was then calculated based on the planting density. Throughout the growth period, the W1, W2, W3, and W4 treatments utilized a total of 25.5 m$^3$, 29.22 m$^3$, 31.47 m$^3$, and 32.48 m$^3$ of water, respectively. Figure 5 illustrates the cumulative daily water consumption under different treatments, indicating an increasing trend in the total water consumption with increases in the irrigation threshold. The CK treatment, irrigated with 26.6 m$^3$ of water using the local water-saving irrigation system, was used as the control. Compared to CK, the W1 treatment achieved 4.85% water savings.

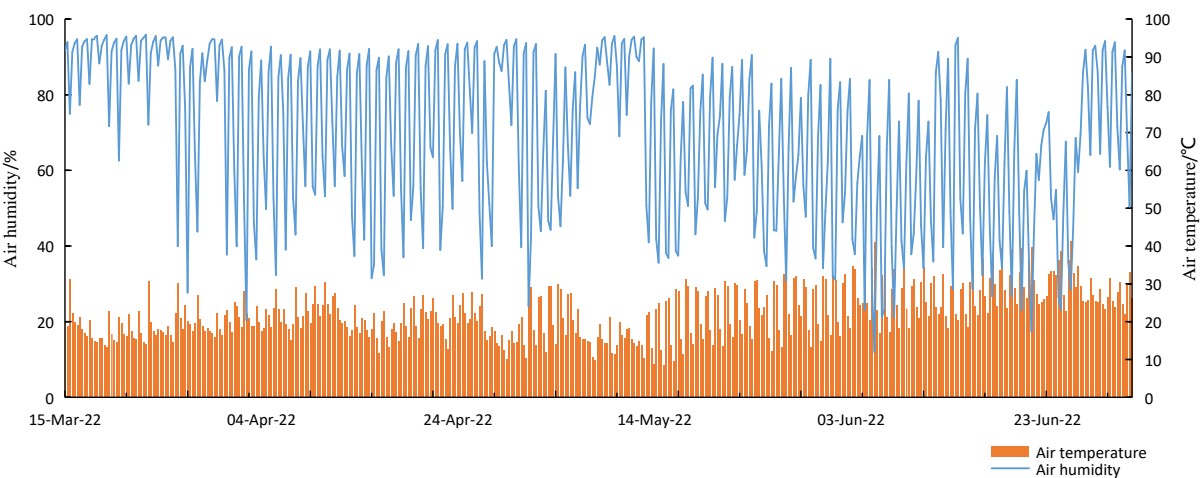

**Figure 4.** Air temperature and humidity in solar greenhouse.

**Figure 5.** Cumulative daily water usage curves for different treatments.

2.4.2. Tomato Yield and Quality

During the harvest period, one or two panicles of fruit were randomly selected from three plants, and the fruit weight was measured using a balance with an accuracy of 0.01 g to determine the yield per plant (in kg/plant). Five fruits with uniform size and hardness were randomly selected from each treatment for analysis. The content of vitamin C (VC in mg/100 g) was determined using the 2,6-dichlorophenol indophenol sodium titration method [26]; the soluble sugar (in mg/100 g) was measured using the Rhein–Enon's method [17]; the lycopene (in mg/100 g) was measured using the spectrophotometer method [27]; the titratable acid (%), which was determined as the major acid in tomato, was calculated using the NaOH titration method [28]; and the total soluble solids (TSS, %) were measured using the PAL-1 digital sweetness tester (Atago, Tokyo, Japan). According to the Fang. L (2020), the total soluble solid includes the amount of soluble sugar in the fruit [29]. The sugar–acid ratio was calculated as the value of total soluble solids divided by that of the titratable acid [30]. Each measurement was repeated three times, and the average value was used for further analysis.

2.4.3. N, P, and K Contents of Plants

During the harvest period, the samples were randomly selected from the south, middle, and north of each plot, with a total of three samples. The process was conducted three times in total. The roots were carefully dug, and the above-ground parts were kept intact. The plant samples were then subjected to a controlled drying process at 105 °C to eliminate all the moisture content. Subsequently, the roots, stems, and leaves were heated to 80 °C, while the fruits were heated to 50 °C and dried to a constant mass. Finally, the dry biomass (kg) of each plant sample was precisely weighed using a balance with an accuracy of 0.01 g. The harvested plant samples were ground and thoroughly mixed. Their N content (%) was determined using the Kjeldahl method [31], the P content (%) was measured using a spectrophotometer method [32], and the K content (%) was determined using a flame photometric method [33].

2.4.4. Water Use Efficiency, Nutrient Accumulation, and Use Efficiency

The water use efficiency was calculated based on the yield and water consumption:

$$IWUE_y = Y/ET \tag{2}$$

where $IWUE_y$ is the water use efficiency (kg/m$^3$), $Y$ is the yield per plant (kg/plant), and $ET$ is the water consumption per plant (m$^3$/plant).

The nutrient use efficiency was calculated based on the plant biomass and the N, P, and K contents [34,35]:

$$NUE = \frac{B \times NC}{A} \times 100\% \tag{3}$$

where $NUE$ is the nutrient use efficiency (%), $B$ is the plant biomass (g), $NC$ is the nutrient accumulation (%), and $A$ is the nutrient supply (g/plant).

*2.5. Data Analysis Methods*

The measured tomato data were averaged, and the standard error of the mean (SEM) was included in the graph. One-way analysis of variance (ANOVA) [36] was employed to determine whether differences between the data were significant. Additionally, two-way ANOVA [37] was utilized to investigate whether interactions existed among the two factors, the irrigation, and the nitrogen forms. If the null hypothesis was rejected, Duncan's multiple range test (Duncan's MRT, $p < 0.05$) was utilized as a post-hoc test for variance analysis [38].

The data were subjected to multivariate statistical analysis using IBM SPSS Statistics 23 [39]. Spearman correlations ($p < 0.05$) were calculated for all indicators, and all indicators were normalized and then evaluated using principal component analysis [40] for the tomato yield, quality, water–fertilizer use efficiency. Then, the evaluation values and rankings of

these three dimensions were evaluated in combination with the fuzzy Borda method [41]. The experimental data were organized and plotted using Microsoft Excel 2007 software.

The fuzzy Borda method calculated the membership degree of each evaluation item to the evaluation index score according to Equation (4), determining the ability of the evaluation items to obtain good evaluation results.

$$\mu_{ij} = \frac{x_{ij} - \min_{i}\{x_{ij}\}}{\max_{i}\{x_{ij}\} - \min_{i}\{x_{ij}\}} \times 0.9 + 0.1 \tag{4}$$

where $x_{ij}$ represents the score of the $i$th treatment in the $j$th evaluation method; $\mu_{ij}$ represents membership degree of the $i$th item under the $j$th evaluation method.

The fuzzy number and fuzzy frequency of the $i$th item in the $h$th position were calculated according to Equations (5)–(7).

$$\rho_{hi} = \sum_{j=1}^{n} \delta_{hi}\mu_{ij} \tag{5}$$

$$W_{hi} = \frac{\rho_{hi}}{\sum_{h}\rho_{hi}} \tag{6}$$

where $h$ is the number of evaluation item indicators.

$$\delta_{hi} = \begin{cases} 1, & \text{the } i\text{th item in the } h\text{th position} \\ 0, & \text{others} \end{cases} \tag{7}$$

Converted the ranking into scores based on Equation (8):

$$Q_{hi} = \frac{1}{2}(q - h)(q - h + 1) \tag{8}$$

where $Q_{hi}$ is the score of the $i$th item in the $h$th position, and $q$ is the total number of evaluation indicators.

The fuzzy Borda count of the $i$th term was calculated based on Equation (9):

$$B_i = \sum W_{hi}Q_{hi} \tag{9}$$

## 3. Results

### 3.1. Effect of Different Treatments on Tomato Yield and Quality

Table 4 shows that there were significant differences in the tomato yield under different irrigation levels ($p < 0.05$). The nitrogen fertilizer treatment and the interaction effect of the water and fertilizer had no significant effects on the tomato yield. The highest yield was observed in W3F2, with 72.21%, 97.92%, 44.67%, and 56.16% increases compared to the CKF2, W1F2, W2F2, and W4F2 treatments, respectively, and 9.62% and 34.75% increases compared to W3F1 and W3F3 at the same irrigation level. The yield per plant demonstrated an increasing trend followed by a decreasing trend with the increase in the irrigation lower limit, and the highest yield was achieved under the W3 irrigation level, showing increases of 51.81%, 46.51%, 18.59%, and 25.69% when compared to CK, W1, W2, and W4, respectively. The highest yield was obtained under the F1 treatment, showing increases of 13.08% and 17.91% compared to F2 and F3, respectively.

Different treatments had significant effects on the Vc, soluble sugar, TSS, sugar–acid ratio, and lycopene content of the tomato fruits. Meanwhile, different irrigation and nitrogen forms as well as the water–fertilizer interaction presented effects on all the indicators of tomato quality, which reached highly significant levels ($p < 0.01$). The Vc, soluble sugar, and TSS contents were the highest under the CK1F2 treatment, where the Vc content increased by 31.25%, 10.53%, 13.50%, and 19.98% compared to W1F2, W2F2, W3F2, and W4F2, respectively. The soluble sugar increased by 3.61% to 36.08%, and the TSS increased by 8.00% to 29.50%, respectively.

**Table 4.** Effects of different treatments on tomato yield and quality.

| Irrigation Level | Nitrogen Fertilizer Treatment | Yield per Plant (kg/Plant) | Vc (mg/100 g) | Soluble Sugar (g/100 g) | TSS (%) | Sugar–Acid Ratio | Lycopene (mg/100 g) |
|---|---|---|---|---|---|---|---|
| W1 | F1 | 3.82 ± 0.85 abc | 38.40 ± 1.96 cde | 32.73 ± 1.59 bc | 4.17 ± 0.06 f | 9.70 ± 0.47 bc | 12.20 ± 0.20 cde |
|  | F2 | 2.88 ± 0.64 c | 36.14 ± 1.96 def | 37.60 ± 1.38 a | 5.00 ± 0.17 b | 11.14 ± 0.41 a | 15.20 ± 0.40 b |
|  | F3 | 3.62 ± 0.52 abc | 40.66 ± 0.00 bc | 30.39 ± 0.60 cd | 4.60 ± 0.00 cde | 8.15 ± 0.90 cd | 16.10 ± 0.30 b |
| W2 | F1 | 5.14 ± 1.37 ab | 36.14 ± 1.96 def | 28.24 ± 0.61 d | 4.20 ± 0.100 f | 8.36 ± 0.18 cd | 11.53 ± 0.35 def |
|  | F2 | 3.94 ± 1.69 abc | 42.91 ± 1.96 b | 29.22 ± 1.44 d | 4.50 ± 0.17 de | 7.09 ± 0.28 d | 17.50 ± 0.60 a |
|  | F3 | 3.68 ± 1.31 abc | 35.01 ± 1.96 ef | 30.13 ± 1.00 cd | 4.43 ± 0.06 e | 8.50 ± 0.80 cd | 11.93 ± 0.25 cdef |
| W3 | F1 | 5.20 ± 0.41 ab | 32.75 ± 1.96 fg | 27.76 ± 1.41 d | 4.03 ± 0.06 f | 7.46 ± 1.20 d | 17.57 ± 0.95 a |
|  | F2 | 5.70 ± 1.19 a | 41.79 ± 1.96 bc | 37.68 ± 1.66 a | 4.73 ± 0.06 c | 9.57 ± 0.42 bc | 10.53 ± 1.05 ef |
|  | F3 | 4.23 ± 0.32 abc | 30.49 ± 0.00 g | 26.80 ± 0.65 d | 4.60 ± 0.10 cde | 6.80 ± 0.16 d | 10.53 ± 1.05 ef |
| W4 | F1 | 4.44 ± 1.31 abc | 38.40 ± 1.96 cde | 26.59 ± 0.56 d | 3.07 ± 0.06 g | 5.47 ± 0.46 e | 13.20 ± 0.40 cd |
|  | F2 | 3.65 ± 0.82 abc | 39.53 ± 1.96 bcd | 28.69 ± 2.77 d | 4.17 ± 0.06 f | 8.50 ± 0.82 cd | 13.63 ± 0.55 c |
|  | F3 | 3.96 ± 0.79 abc | 27.10 ± 0.00 h | 17.88 ± 2.17 e | 4.00 ± 0.00 f | 5.30 ± 0.64 e | 13.67 ± 1.25 c |
| CK | F1 | 3.46 ± 0.90 bc | 39.53 ± 1.96 bcd | 37.84 ± 3.08 a | 5.10 ± 0.10 b | 8.41 ± 0.69 cd | 10.37 ± 0.90 f |
|  | F2 | 3.31 ± 1.16 bc | 47.43 ± 0.00 a | 39.04 ± 0.48 a | 5.40 ± 0.10 a | 10.46 ± 0.96 ab | 12.80 ± 0.80 cd |
|  | F3 | 3.20 ± 0.56 bc | 36.14 ± 1.96 def | 33.93 ± 1.24 b | 4.70 ± 0.10 cd | 9.55 ± 0.56 bc | 12.10 ± 0.20 cdef |
| Irrigation level | W1 | 3.44 ± 0.73 b | 38.40 ± 2.40 b | 33.58 ± 3.37 b | 4.59 ± 0.37 b | 9.66 ± 1.40 a | 14.50 ± 1.79 a |
|  | W2 | 4.25 ± 1.44 ab | 38.02 ± 4.07 b | 29.19 ± 1.24 d | 4.38 ± 0.17 c | 7.98 ± 0.80 b | 13.66 ± 2.91 b |
|  | W3 | 5.04 ± 0.92 a | 35.01 ± 5.36 c | 30.75 ± 5.34 c | 4.46 ± 0.33 c | 7.94 ± 1.37 b | 13.07 ± 3.49 b |
|  | W4 | 4.01 ± 0.93 b | 35.01 ± 6.11 c | 24.39 ± 5.27 e | 3.74 ± 0.52 d | 6.42 ± 1.66 c | 13.50 ± 0.75 b |
|  | CK | 3.32 ± 0.80 b | 41.03 ± 5.21 a | 36.94 ± 2.86 a | 5.07 ± 0.32 a | 9.47 ± 1.11 a | 11.76 ± 1.25 c |
| Nitrogen fertilizer treatment | F1 | 4.41 ± 1.13 a | 37.04 ± 2.99 b | 30.63 ± 4.55 b | 4.11 ± 0.67 c | 7.88 ± 1.54 b | 12.97 ± 2.62 b |
|  | F2 | 3.90 ± 1.40 a | 41.56 ± 4.14 a | 34.45 ± 4.89 a | 4.76 ± 0.45 a | 9.35 ± 1.58 a | 13.93 ± 2.49 a |
|  | F3 | 3.74 ± 0.75 a | 33.88 ± 4.96 c | 27.83 ± 5.75 c | 4.47 ± 0.26 b | 7.66 ± 1.62 b | 12.98 ± 1.93 b |
| *p* | W | 0.007 | 0.000 | 0.000 | 0.000 | 0.000 | 0.000 |
|  | F | 0.171 | 0.000 | 0.000 | 0.000 | 0.000 | 0.001 |
|  | W*F | 0.67 | 0.000 | 0.000 | 0.000 | 0.000 | 0.000 |

Note(s): Different lowercase letters in the same column indicate significant differences among treatments ($p < 0.05$). Values with the same letters are not significantly different. The multiplication sign represents there is an interaction effect.

With respect to the irrigation level, the increase in the irrigation lower limit led to an increase in the irrigation water volume, followed by a decreasing trend in the Vc, soluble sugar, and TSS. The highest sugar–acid ratio was observed in W1F2 (11.14), followed by CKF2 (10.46), ranking as W1 > CK > W2 > W3 > W4 under different irrigation treatments, which the appropriate sugar–acid ratio is 6.9 to 11 [42]. The lycopene presented a significant difference among the various treatments. The highest lycopene could be witnessed at W3F1, and it was higher than that of the lowest treatment 69.43%. F2 was significantly better than F1 and F3 in all indicators of tomato quality under different nitrogen forms ($p < 0.01$).

### 3.2. Effects of Different Treatments on Water and Fertilizer Use Efficiency of Tomato

Figure 6 presents the significant differences in the nutrient accumulation and use efficiencies in tomato under the different treatments. W2F2 demonstrated the highest accumulation and use rates for all nutrients compared to the other treatments. The N, $P_2O_5$, and $K_2O$ accumulations in W2F2 were 10.76 g/plant, 1.80 g/plant, and 15.86 g/plant, and the use efficiencies were 74.58%, 31.83%, and 77.06%, respectively. Compared to the other treatments, W2F2 showed significant increases in the N content (12.46% to 80.16%), the N use rate (12.47% to 80.19%), the $P_2O_5$ content (7.80% to 125.01%), the $P_2O_5$ use rate (7.79% to 85.81%), the $K_2O$ content (12.24% to 128.20%), and the $K_2O$ use rate (12.20% to 128.06%).

Highly significant differences were observed in the nutrient accumulation of tomato under different irrigation treatments ($p < 0.01$). Furthermore, there were no significant differences in the accumulation of N and K, but significant differences were found in the accumulation of P under the three nitrogen fertilizer treatments ($p < 0.05$). There were significant differences in the accumulation of N, P, and K under the water–fertilizer interaction ($p < 0.01$). The effects of different irrigation treatments, nitrogen fertilizer treatments, and the interaction effects of water and fertilizer on the use efficiency of N, P, and K reached a highly significant level ($p < 0.01$).

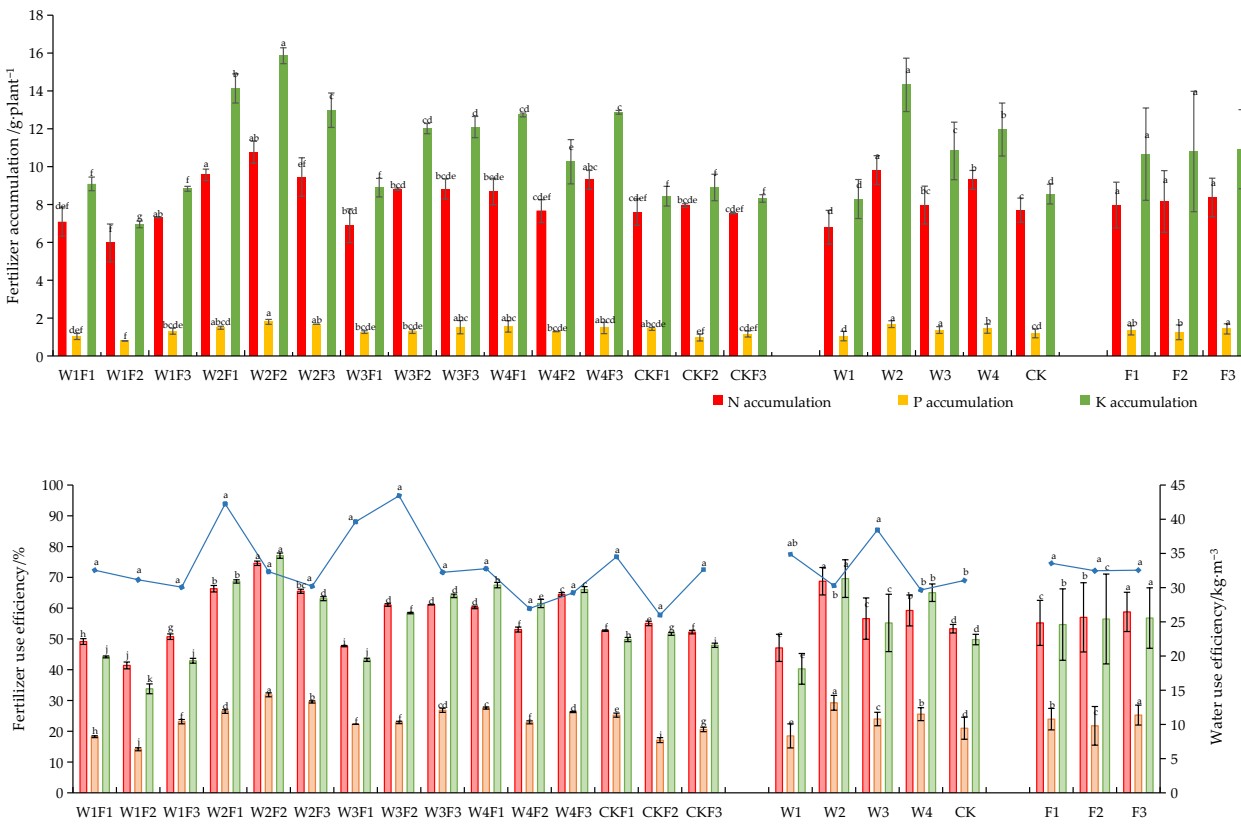

**Figure 6.** Effect of different treatments on irrigation water use effciency and nutrient use efficiency of tomatoes. Different lowercase letters in the same column indicate significant differences among treatments ($p < 0.05$). Values with the same letters are not significantly different. The multiplication sign represents there is an interaction effect. The detailed data can be found at Appendix A.

No significant differences were observed in the water use efficiency of tomato among the 15 groups under different water and fertilizer treatments. Furthermore, there were no significant differences in the water use efficiency of tomato under the same irrigation regime with different nitrogen forms and water–fertilizer interaction effects. However, different irrigation treatments at the same fertilization level had a significant effect on the water use efficiency of tomato, with $p < 0.05$. The water use efficiency of W3F2 was the highest at 43.430 kg/m$^3$, but did not reach a significant level. Under the same nitrogen fertilizer treatment, it was 67.04%, 39.44%, 34.29%, and 61.17% higher than that of CKF2, W1F2, W2F2, and W4F2 with different irrigation levels. Under the same irrigation level, it was 9.59% and 34.72% higher than that of W3F1 and W3F3 under different nitrogen fertilizer treatments.

### 3.3. Comprehensive Evaluation of the Effect of Different Water and Fertilizer Treatments
#### 3.3.1. Correlation between Indicators

The yield per plant, Vc, soluble sugar, TSS, sugar–acid ratio, lycopene, water use efficiency, N use efficiency, P$_2$O$_5$ use efficiency, and K$_2$O use efficiency are represented by U1~U10. The correlations among the indicators are tabulated in Table 5. According to the results, the yield per plant had a negative correlation with the soluble sugar, TSS, and sugar–acid ratio, while showing a highly significant positive correlation with the water use efficiency. Additionally, Vc was found to be positively correlated with the soluble sugar, and the soluble sugar had a positive correlation with the TSS and sugar–acid ratio. On the other hand, the soluble sugar had a negative correlation with the P use efficiency and K use efficiency. The TSS showed a positive correlation with the sugar–acid ratio. The sugar–acid

ratio was highly negatively correlated with the P use efficiency and negatively correlated with the K use efficiency. There was no significant correlation between the lycopene and the other indicators. There were highly significant positive correlations among the N use efficiency, $P_2O_5$ use efficiency, and $K_2O$ use efficiency.

**Table 5.** Spearman's correlation between indicators.

| No. | $U_1$ | $U_2$ | $U_3$ | $U_4$ | $U_5$ | $U_6$ | $U_7$ | $U_8$ | $U_9$ | $U_{10}$ |
|---|---|---|---|---|---|---|---|---|---|---|
| $U_1$ | 1 | | | | | | | | | |
| $U_2$ | −0.219 | 1 | | | | | | | | |
| $U_3$ | −0.579 * | 0.614 * | 1 | | | | | | | |
| $U_4$ | −0.558 * | 0.453 | 0.852 ** | 1 | | | | | | |
| $U_5$ | −0.527 * | 0.371 | 0.833 ** | 0.603 * | 1 | | | | | |
| $U_6$ | −0.075 | 0.032 | −0.293 | −0.37 | −0.204 | 1 | | | | |
| $U_7$ | 0.557 * | −0.061 | 0.007 | −0.057 | −0.077 | −0.35 | 1 | | | |
| $U_8$ | 0.429 | −0.005 | −0.361 | −0.166 | −0.45 | −0.281 | 0.007 | 1 | | |
| $U_9$ | 0.432 | −0.147 | −0.579 * | −0.386 | −0.745 ** | −0.118 | 0.089 | 0.789 ** | 1 | |
| $U_{10}$ | 0.493 | −0.144 | −0.536 * | −0.376 | −0.570 * | −0.195 | 0.064 | 0.929 ** | 0.796 ** | 1 |

Note(s): "*" indicates a significant correlation at the $p < 0.05$ level, and "**" indicates a highly significant correlation at the $p < 0.01$ level.

The results demonstrate that the indicator data reflected both overlapping and distinct information. To better evaluate the comprehensive effects of the different irrigation and fertilization systems, a comprehensive multi-item evaluation was necessary.

3.3.2. Analysis and Evaluation of Tomato Yield, Quality, and Water–Fertilizer Use Efficiency

Principal component analysis was used to analyze the indicators of tomato quality and water–fertilizer use efficiency, respectively. The Vc ($U_2$), soluble sugar ($U_3$), TSS ($U_4$), sugar–acid ratio ($U_5$), lycopene ($U_6$), water use efficiency ($U_7$), N use efficiency ($U_8$), $P_2O_5$ use efficiency ($U_9$), and $K_2O$ use efficiency ($U_{10}$) were normalized and then subjected to principal component analysis, respectively.

The contributions of the eigenvalues of the tomato quality indicators are shown in Table 6, and two principal components were extracted. The variance interpretation rates of the two principal components were 59.965% and 20.757%, respectively, and the cumulative variance interpretation rates were 80.723%. The principal component expression is:

$$Y_1 = 0.413U_2 + 0.556U_3 + 0.479U_4 + 0.520U_5 - 0.141U_6 \quad (10)$$

$$Y_2 = 0.401U_2 - 0.082U_4 + 0.003U_5 + 0.912U_6 \quad (11)$$

**Table 6.** Principal component variance interpretation of tomato quality indicators under each treatment.

| No. | Eigenvalue | Variance Interpretation Rate (%) | Cumulative Variance Interpretation Rate (%) |
|---|---|---|---|
| $U_2$ | 2.998 | 59.965 | 59.965 |
| $U_3$ | 1.038 | 20.757 | 80.723 |
| $U_4$ | 0.582 | 11.631 | 92.354 |
| $U_5$ | 0.297 | 5.932 | 98.286 |
| $U_6$ | 0.086 | 1.714 | 100.000 |

The greatest influences on the first and second principal components were the Vc and soluble sugar, respectively, and the first two principal components were used as the evaluation indicators of the tomato quality, with the integrated evaluation equation of

$$Y = 0.743Y_1 + 0.257Y_2 \quad (12)$$

The contributions of the eigenvalues of the water and fertilizer use efficiency indicators are presented in Table 7, and one principal component was extracted. The cumulative contribution of this principal component was 69.317%, which can replace the original four variables to evaluate the water and nutrient use efficiencies under different treatments. The principal component expression is:

$$Z_1 = 0.109U_7 + 0.583U_8 + 0.561U_9 + 0.578U_{10} \quad (13)$$

**Table 7.** Principal component variance interpretation of water and fertilizer use efficiency under each treatment.

| No. | Eigenvalue | Variance Interpretation Rate (%) | Cumulative Variance Interpretation Rate (%) |
|---|---|---|---|
| $U_7$ | 2.773 | 69.316 | 69.316 |
| $U_8$ | 0.984 | 24.591 | 93.906 |
| $U_9$ | 0.187 | 4.685 | 98.591 |
| $U_{10}$ | 0.056 | 1.409 | 100.000 |

The comprehensive score was calculated according to the first principal component:

$$Z = Z_1 \quad (14)$$

The comprehensive score and ranking of principal components of the tomato quality and water and fertilizer use efficiency were obtained according to Equations (12) and (14). Table 8. lists the rankings and scores reflecting the three dimensions of tomatoes.

**Table 8.** Rankings and scores reflecting the three dimensions of tomato.

| Irrigation Level | Nitrogen Fertilizer Treatment | Tomato Quality Indicator | | Water and Fertilizer Use Efficiency Indicator | | Yield Indicator | |
|---|---|---|---|---|---|---|---|
| | | Score | Ranking | Score | Ranking | Yield | Ranking |
| W1 | F1 | 0.302 | 7 | −1.735 | 14 | 3.823 | 8 |
| | F2 | 1.470 | 2 | −3.258 | 15 | 2.882 | 15 |
| | F3 | 0.428 | 6 | −1.170 | 11 | 3.618 | 11 |
| W2 | F1 | −0.538 | 11 | 1.745 | 2 | 5.144 | 3 |
| | F2 | 0.297 | 8 | 3.122 | 1 | 3.937 | 7 |
| | F3 | −0.299 | 10 | 1.533 | 3 | 3.678 | 10 |
| W3 | F1 | −0.828 | 12 | −1.256 | 12 | 5.196 | 2 |
| | F2 | 1.154 | 3 | 0.511 | 7 | 5.695 | 1 |
| | F3 | −1.275 | 13 | 1.012 | 6 | 4.227 | 5 |
| W4 | F1 | −1.731 | 14 | 1.212 | 4 | 4.435 | 4 |
| | F2 | −0.101 | 9 | −0.224 | 8 | 3.647 | 9 |
| | F3 | −2.738 | 15 | 1.199 | 5 | 3.958 | 6 |
| CK | F1 | 0.929 | 4 | −0.365 | 9 | 3.458 | 12 |
| | F2 | 2.444 | 1 | −1.257 | 13 | 3.309 | 13 |
| | F3 | 0.485 | 5 | −1.071 | 10 | 3.195 | 14 |

### 3.3.3. Comprehensive Evaluation of the Effect of Different Water and Fertilizer Treatments Using Fuzzy Borda

The evaluation information of the three dimensions of yield, quality, and water–fertilizer use efficiency was evaluated in combination using the fuzzy Borda method to obtain the comprehensive score and ranking. The results are shown in Figure 7, where they are ranked by area size. The W3F2 scored 78.208, ranking first, and indicating that under the treatment with 70%FC~90%FC as the irrigation condition and with nitrate nitrogen fertilizer, the multi-dimensional optimal comprehensive evaluation of the tomato yield, quality, and water–fertilizer use efficiency can be achieved.

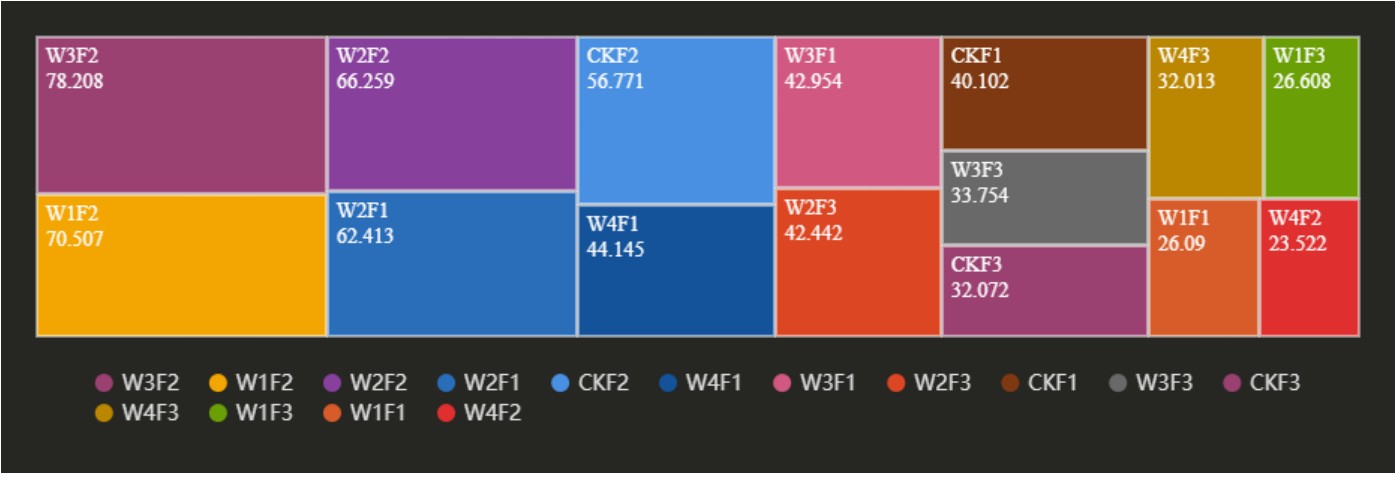

**Figure 7.** The comprehensive evaluation score and ranking using the fuzzy Borda method.

## 4. Discussion

### 4.1. Relationships among Tomato Yield, Quality Indicators, and Water and Fertilizer Use Efficiency Indicators

The optimal regression equation was obtained by a stepwise regression method through a path analysis, with tomato yield ($U_1$) as the dependent variable and the indicators of tomato quality and water and fertilizer use efficiency ($U_1 \sim U_{10}$) as independent variables: $U_1 = 1.039 - 0.156U_5 + 0.129U_7$ ($R^2 = 0.782$). This implies that the sugar–acid ratio and water use efficiency could explain 78.3% of the variation in the yield per plant. The path analysis showed that the sugar–acid ratio had a significant negative correlation with the yield, while the water use efficiency had a significant positive correlation with the yield, with a path coefficient of 0.947. These results suggest that improving the tomato water use efficiency can be a key factor in increasing tomato yield.

### 4.2. Effects of Irrigation Lower Limit and Nitrogen Forms and Water-Fertilizer Interactions on Tomato Yield and Quality

Soil moisture and fertility are important factors that significantly affect crop yield and quality. Research on the irrigation and fertilization of tomatoes under facility growing conditions has been widely conducted by scholars both domestically and abroad [43,44]. Many studies have demonstrated that irrigation, fertilization, and intercropping practices generate certain effects on crop yield and quality [45–47].

In this study, significant differences were observed in the yield among 15 groups of treatments with different irrigation and fertilization regimes. Additionally, highly significant yield differences were observed among W1, W2, W3, W4, and CK under different irrigation treatments ($p < 0.01$). However, no significant differences in yield were observed among F1, F2, and F3 under different nitrogen forms, and no significant effects of the water–fertilizer interaction were observed on the tomato yield.

The treatment that resulted in the highest yield per plant was W3F2, with a yield of 5.695 kg. The variation in the yield among treatments followed a trend of increasing

and then decreasing with increases in the irrigation lower limit. This can be interpreted that the gradual increase in the irrigation starting conditions, from 50%FC to 80%FC for each treatment W1 to W4, resulted in a gradual increase in the irrigation water volume throughout the entire growth period of the tomato. However, as the lower limit continued to increase, the air permeability of the soil decreased, which inhibited the ability of the root system to absorb nutrients, ultimately resulting in a decrease in the fruit yield [48].

In this study, the irrigation and nitrogen forms affected all indicators of tomato quality at highly significant levels ($p < 0.01$), while the water–fertilizer interaction effects were also found to have highly significant effects on the tomato quality ($p < 0.01$). Irrigation and fertilization had profound impacts on the tomato fruit quality through different mechanisms. Irrigation influenced the water content of the fruit, thereby altering its nutrient content and quality. The nitrogen form affected the activity of key enzymes involved in the organic acid metabolism, such as phosphoenolpyruvate carboxylase (PEPCase), as well as nitrogen metabolism enzymes, such as nitrate reductase (NR) and glutamine synthetase (GS) in the fruit, which in turn modulated the sugar–acid ratio, thereby impacting the flavor and quality of the tomato [49].

The relationship model between the quality comprehensive score ($Y$) and the tomato yield ($U_1$) was determined through a quadratic curve fitting method, as shown in Equation (15).

$$Y = 26.35 - 12.18U_1 + 1.35U_1^2 \qquad (15)$$

As depicted in Figure 8, the quality score shows a decreasing and then increasing trend with an increasing yield. The lowest quality score was observed when the yield per plant increased to 4.51 kg/plant, after which the quality score gradually increased with further increases in the yield. The reason for this situation may have been that there was a negative correlation between the quality score and the yield before the yield per plant increased to 4.51 kg. With the increase in the irrigation volume and appropriate fertilization, both the yield and quality improved simultaneously.

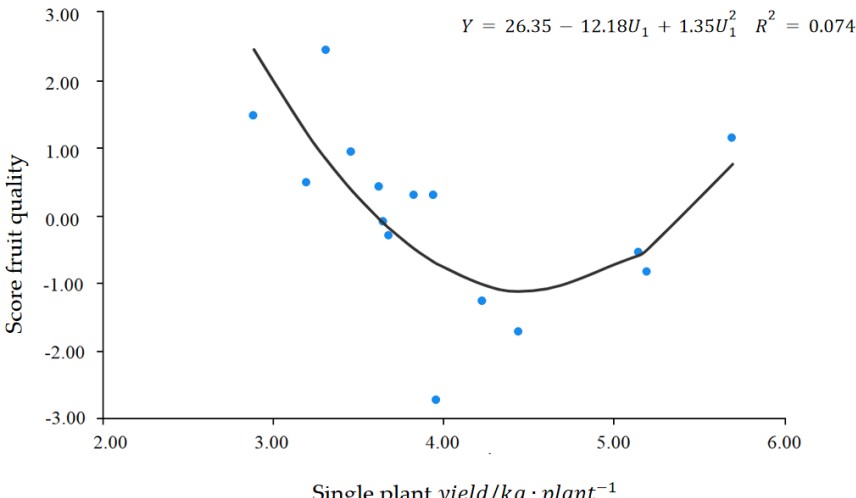

**Figure 8.** Quadratic regression curve of quality compressive score and yield. The blue dots represent the final evaluation including quality scores and yields for all treatments.

### 4.3. Effects of Irrigation Lower Limit and Nitrogen Forms and Water-Fertilizer Interactions on Water and Fertilizer Use Efficiency of Tomato

The results show that the different irrigation treatments, under the same amount of fertilization, had significantly different effects on the tomato water use efficiency ($p < 0.05$). The highest water use efficiency, 38.432 kg/m$^3$, was observed when the irrigation lower limit was set at 70%FC. However, no clear trend was observed between the water use efficiency and the irrigation lower limit. This was because different water and nitrogen supply patterns affected the water–fertilizer use efficiency of plants [50,51]. Wang et al.

concluded that reduced soil water regimes under N fertigation caused the partial closure of the stomata via the decreased plant water status and intensified root-to-shoot abscisic acid (ABA) signaling, resulting in an improved intrinsic water use efficiency.

The results of this study indicate that the effects of irrigation schemes, nitrogen forms, and their interaction on the utilization efficiency of N, P, and K were highly significant. The principal component analysis revealed that the highest comprehensive score for water and fertilizer use efficiency (W2F2) was achieved when the irrigation lower limit was set to 60%FC. Studies have shown that there is a positive correlation between the appropriate application of nitrate nitrogen and yield and nutrient element use efficiency in crops such as maize [52] and cotton [53]. The results of this study indicate that under water–fertilizer interaction, an appropriate amount of water and nitrogen can greatly enhance plant and fruit growth, resulting in improved nutrient element use efficiency while maintaining a certain level of fertilization.

In future research, the performance of water and fertilizer use efficiency on various tomato cultivars should be discussed, and the various requirements of irrigation volume in different growth periods of tomato should be explored.

## 5. Conclusions

The water–fertilizer interaction has highly significant impacts on the quality indicators of tomato, such as the Vc, soluble sugar, TSS, sugar–acid ratio, and lycopene content. Furthermore, the interaction also significantly influences the accumulation and use efficiency of the N, P, and K in tomato. These findings indicate that the improvement in the water use efficiency of tomato plays a crucial role in increasing tomato yield. The optimal irrigation and fertilization scheme for the spring crop cultivation of facility tomato in this region is determined as follows: The soil moisture should be controlled at 70%FC from the fruit setting to the harvest period. At the same time, the application of nitrate nitrogen fertilizer could achieve the best effects on the tomato yield, quality, and water and fertilizer use efficiency.

**Author Contributions:** Conceptualization, Y.H. and G.-Y.T.; methodology, Y.-R.Y.; software, Y.-R.D.; validation, Y.-F.K.; formal analysis, Y.H. and G.-Y.T.; investigation, J.-X.Y.; resources, J.-X.H.; data curation, Y.H.; writing—original draft preparation, Y.H.; writing—review and editing, Y.-R.Y.; visualization, Y.-R.D.; supervision, J.-X.Y.; project administration, G.-Y.T.; funding acquisition, G.-Y.T. All authors have read and agreed to the published version of the manuscript.

**Funding:** This research was funded by Hebei Province Key R&D Program Project (21327410D), Hebei Province Key R&D Program Project (22327401D), and Shijiazhuang Municipal Science and Technology Research and Development Program (221490072A).

**Institutional Review Board Statement:** Not applicable.

**Informed Consent Statement:** Not applicable.

**Data Availability Statement:** Not applicable.

**Conflicts of Interest:** The authors declare no conflict of interest.

# Appendix A

**Table A1.** Effects of different treatments on IWUE and NUE of tomato.

| Treatment | N Accumulation (g/plant) | P Accumulation (g/plant) | K Accumulation (g/plant) | N Use Efficiency (%) | P Use Efficiency (%) | K Use Efficiency (%) | Water Use Efficiency (kg/·m³) |
|---|---|---|---|---|---|---|---|
| W1F1 | 7.09 ± 0.77 [def] | 1.03 ± 0.16 [def] | 9.09 ± 0.36 [f] | 49.15 ± 0.94 [h] | 18.24 ± 0.30 [h] | 44.16 ± 0.32 [j] | 32.55 ± 8.51 [a] |
| W1F2 | 5.97 ± 1.00 [f] | 0.80 ± 0.00 [f] | 6.95 ± 0.18 [g] | 41.39 ± 1.11 [j] | 14.15 ± 0.49 [j] | 33.79 ± 1.60 [k] | 31.15 ± 10.91 [a] |
| W1F3 | 7.32 ± 0.02 [def] | 1.31 ± 0.16 [bcde] | 8.83 ± 0.13 [f] | 50.73 ± 0.90 [g] | 23.11 ± 0.70 [f] | 42.91 ± 0.79 [j] | 30.07 ± 5.24 [a] |
| W2F1 | 9.57 ± 0.30 [ab] | 1.49 ± 0.08 [abcd] | 14.13 ± 0.77 [b] | 66.31 ± 1.05 [b] | 26.40 ± 0.58 [d] | 68.68 ± 0.54 [b] | 42.25 ± 11.22 [a] |
| W2F2 | 10.76 ± 0.58 [a] | 1.80 ± 0.13 [a] | 15.86 ± 0.42 [a] | 74.58 ± 0.71 [a] | 31.83 ± 0.63 [a] | 77.06 ± 0.86 [a] | 32.34 ± 13.89 [a] |
| W2F3 | 9.45 ± 1.02 [ab] | 1.67 ± 0.00 [ab] | 12.98 ± 0.91 [c] | 65.49 ± 0.67 [bc] | 29.53 ± 0.37 [b] | 63.07 ± 0.65 [d] | 30.21 ± 10.76 [a] |
| W3F1 | 6.88 ± 0.89 [ef] | 1.26 ± 0.07 [bcde] | 8.89 ± 0.49 [f] | 47.66 ± 0.21 [i] | 22.26 ± 0.00 [f] | 43.21 ± 0.52 [j] | 39.63 ± 3.15 [a] |
| W3F2 | 8.81 ± 0.06 [bcd] | 1.29 ± 0.10 [bcde] | 12.01 ± 0.27 [cd] | 61.02 ± 0.43 [d] | 22.86 ± 0.39 [f] | 58.39 ± 0.23 [f] | 43.43 ± 9.08 [a] |
| W3F3 | 8.82 ± 0.53 [bcde] | 1.52 ± 0.35 [abc] | 12.09 ± 0.56 [d] | 61.14 ± 0.06 [d] | 26.83 ± 0.68 [cd] | 63.99 ± 0.57 [d] | 32.24 ± 2.46 [a] |
| W4F1 | 8.69 ± 0.70 [bcde] | 1.56 ± 0.30 [abc] | 12.73 ± 0.10 [c] | 60.24 ± 0.42 [d] | 27.54 ± 0.34 [c] | 67.49 ± 0.88 [b] | 32.77 ± 9.66 [a] |
| W4F2 | 7.65 ± 0.60 [cdef] | 1.29 ± 0.01 [bcde] | 10.26 ± 1.17 [e] | 53.04 ± 0.80 [f] | 22.90 ± 0.49 [f] | 61.52 ± 1.34 [e] | 26.95 ± 6.06 [a] |
| W4F3 | 9.30 ± 0.50 [abc] | 1.48 ± 0.30 [abcd] | 12.87 ± 0.10 [c] | 64.46 ± 0.59 [c] | 26.25 ± 0.18 [d] | 66.05 ± 0.96 [c] | 29.25 ± 5.84 [a] |
| CKF1 | 7.60 ± 0.70 [cdef] | 1.43 ± 0.08 [abcde] | 8.44 ± 0.52 [f] | 52.69 ± 0.22 [f] | 25.25 ± 0.66 [e] | 49.80 ± 0.68 [h] | 34.50 ± 7.62 [a] |
| CKF2 | 7.94 ± 0.10 [bcde] | 0.97 ± 0.18 [ef] | 8.90 ± 0.70 [f] | 55.04 ± 0.74 [e] | 17.13 ± 0.80 [i] | 51.65 ± 0.50 [g] | 26.00 ± 5.76 [a] |
| CKF3 | 7.54 ± 0.03 [cdef] | 1.16 ± 0.16 [cdef] | 8.32 ± 0.20 [f] | 52.22 ± 0.57 [f] | 20.60 ± 0.73 [g] | 47.94 ± 0.69 [i] | 32.64 ± 4.66 [a] |
| W1 | 6.80 ± 0.89 [d] | 1.05 ± 0.25 [d] | 8.29 ± 1.03 [d] | 47.09 ± 4.41 [e] | 18.50 ± 3.91 [e] | 40.28 ± 4.99 [e] | 34.87 ± 5.34 [ab] |
| W2 | 9.82 ± 0.76 [a] | 1.68 ± 0.18 [a] | 14.32 ± 1.41 [a] | 68.75 ± 4.46 [a] | 29.25 ± 2.40 [a] | 69.60 ± 6.12 [a] | 30.30 ± 6.04 [b] |
| W3 | 7.97 ± 1.00 [bc] | 1.37 ± 0.23 [bc] | 10.83 ± 1.52 [c] | 56.60 ± 6.72 [c] | 23.98 ± 2.19 [c] | 55.20 ± 9.32 [c] | 38.43 ± 6.99 [a] |
| W4 | 9.30 ± 0.50 [b] | 1.44 ± 0.24 [b] | 11.96 ± 1.40 [b] | 59.25 ± 5.03 [b] | 25.56 ± 2.1 [b] | 65.02 ± 2.85 [b] | 29.66 ± 6.89 [b] |
| CK | 7.70 ± 0.64 [c] | 1.19 ± 0.23 [cd] | 8.55 ± 0.52 [d] | 53.32 ± 1.39 [d] | 20.99 ± 3.58 [d] | 49.80 ± 1.70 [d] | 31.05 ± 6.57 [b] |
| F1 | 7.97 ± 1.21 [a] | 1.35 ± 0.24 [ab] | 10.66 ± 2.44 [a] | 55.21 ± 7.32 [c] | 23.94 ± 3.49 [b] | 54.67 ± 11.60 [b] | 33.56 ± 6.09 [a] |
| F2 | 8.16 ± 1.63 [a] | 1.25 ± 0.39 [b] | 10.80 ± 3.18 [a] | 57.01 ± 11.25 [b] | 21.77 ± 6.28 [c] | 56.48 ± 14.60 [a] | 32.47 ± 8.57 [a] |
| F3 | 8.37 ± 1.02 [a] | 1.43 ± 0.27 [a] | 10.92 ± 2.09 [a] | 58.78 ± 6.38 [a] | 25.26 ± 3.24 [a] | 56.79 ± 9.82 [a] | 32.55 ± 6.40 [a] |
| W | 0.000 | 0.000 | 0.000 | 0.000 | 0.000 | 0.000 | 0.029 |
| F | 0.278 | 0.021 | 0.435 | 0.000 | 0.000 | 0.000 | 0.871 |
| W*F | 0.003 | 0.003 | 0.000 | 0.000 | 0.000 | 0.000 | 0.249 |

Note(s): Different lowercase letters in the same column indicate significant differences among treatments ($p < 0.05$).

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
