# Peer review of "Interaction of the Coupled Effects of Irrigation Mode and Nitrogen Fertilizer Format on Tomato Production"

_water, doi:10.3390/w15081546_

Round 1
Reviewer 1 Report
Interaction of the Coupled Effects of Irrigation Mode and Nitrogen Fertilizer Format on Tomato Production
General comments:
The Coupled Effects of Irrigation Scheduling and Nitrogen Fertilization already published for many crops like
- wheat (Zain et al. 2021 at https://doi.org/10.3390/su13052742),
- jujube (Dai et al. 2019 at https://doi.org/10.1016/j.agwat.2018.09.035)
- tomato (Wu et al. 2022 at DOI: 10.1016/j.agwat.2021.107401
So, this topic is already handled by several researchers and thousands published articles on nitrogen form son tomato production!!
Comments in details:
- Line 70: “Additionally, the interaction between different nitrogen forms and irrigation has been noted in literatures.” Add some refs. please
- Table 1: Available Nitrogen (54.83 mg/g), is that right unit??
I am sure not correct, which equal 54 830 ppm and it is impossible?
Even in ppm is high and that means soil does not need to add any N-fertilizers?
- Table 4: this expression “N absorption efficiency (%)” is not right, please change into “N use efficiency (%)”
- the authors must convert these treatments of irrigation into amount of irrigation water like the control as below:
“The control treatment (CK) was irrigated once every 7-14 days during the flowering and fruiting periods with 80~120 m3 /hm2 of irrigation water each time,”
- the main absent parameter is the economic evaluation using costs of irrigation and fertilization, and energy etc.?
Major revision needed
Reviewer 2 Report
The work is novel in its intensity of detail for a combined study. And it is useful in showing the form of N nutrition is not important under the test conditions. Obviously that leads to more questions: cultivar variability soil type?
The methods section requires more detail and very few references are provided for the assays involved.
There are sticky notes showing where for this reviewer the meaning were unclear plus suggestions for editing spellings etc
I am not clear on the replications for the study

Round 2
Reviewer 1 Report
Many thanks for your corrections!
Please check the introduction section again, you can see “[Error! Reference source not found.].
Please correct!
“Tomato planting effect is a comprehensive concept that includes not only fruit yield 77 and quality directly related to economic benefits, but also water and fertilizer utilization 78 efficiency related to ecological benefits. It is the sum of interactions between different in- 79 dividual attributes [Error! Reference source not found.]. And the fruit quality is also gen- 80 erally classified as taste (total soluble solids, sugar and acid), nutritional (lycopene and 81 vitamin C) [17]. Water and fertilizer utilization efficiency include productivity of water 82 consumption and partial factor productivity of fertilizer application [Error! Reference 83 source not found.].
The authors did not answer this question:
Even in ppm is high and that means soil does not need to add any N-fertilizers?
Why they added N, although the soil content is enough?
How can know this impact of added N back to added or existed N in soil?
Reviewer 2 Report
please look through your changes requires editing in places still
there are places that discuss error in references do not understand
still do not understand some of the terminology
